# Mechanisms Underlying Dichotomous Procoagulant COAT Platelet Generation—A Conceptual Review Summarizing Current Knowledge

**DOI:** 10.3390/ijms23052536

**Published:** 2022-02-25

**Authors:** Lucas Veuthey, Alessandro Aliotta, Debora Bertaggia Calderara, Cindy Pereira Portela, Lorenzo Alberio

**Affiliations:** Hemostasis and Platelet Research Laboratory, Division of Hematology and Central Hematology Laboratory, Lausanne University Hospital (CHUV) and University of Lausanne (UNIL), CH-1010 Lausanne, Switzerland; lucas.veuthey@chuv.ch (L.V.); alessandro.aliotta@chuv.ch (A.A.); debora.bertaggia-calderara@chuv.ch (D.B.C.); cindy.pereira-portela@chuv.ch (C.P.P.)

**Keywords:** procoagulant platelets, COAT, signaling, regulation, heterogeneity, hemostasis

## Abstract

Procoagulant platelets are a subtype of activated platelets that sustains thrombin generation in order to consolidate the clot and stop bleeding. This aspect of platelet activation is gaining more and more recognition and interest. In fact, next to aggregating platelets, procoagulant platelets are key regulators of thrombus formation. Imbalance of both subpopulations can lead to undesired thrombotic or bleeding events. COAT platelets derive from a common pro-aggregatory phenotype in cells capable of accumulating enough cytosolic calcium to trigger specific pathways that mediate the loss of their aggregating properties and the development of new adhesive and procoagulant characteristics. Complex cascades of signaling events are involved and this may explain why an inter-individual variability exists in procoagulant potential. Nowadays, we know the key agonists and mediators underlying the generation of a procoagulant platelet response. However, we still lack insight into the actual mechanisms controlling this dichotomous pattern (i.e., procoagulant versus aggregating phenotype). In this review, we describe the phenotypic characteristics of procoagulant COAT platelets, we detail the current knowledge on the mechanisms of the procoagulant response, and discuss possible drivers of this dichotomous diversification, in particular addressing the impact of the platelet environment during in vivo thrombus formation.

## 1. Introduction

Procoagulant platelets are a subpopulation of platelets appearing upon strong activation, which localize and enhance thrombin generation at sites of vascular injury by expressing negatively charged phospholipids and sustaining the formation of the tenase and prothrombinase complexes on their surface [1]. Eventually, procoagulant platelets promote the deposition of fibrin in order to stabilize and limit the primary platelet plug, which is composed of aggregating platelets, the main platelet subpopulation [2]. Hence, balance between platelet aggregation and procoagulant activity in the growing thrombus is a key element of hemostasis [3,4,5]. The time-dependent appearance and role of procoagulant platelets are shown in Figure 1.

This balance seems to be altered in bleeding diatheses [6,7] and hemorrhagic strokes [8,9] where a low procoagulant potential has been observed. On the other hand, in ischemic strokes [10,11] and in conditions as diverse as COVID-19 [12] and coronary artery disease [13], a high procoagulant potential was associated with a worse outcome. The clinical implications of procoagulant COAT platelets are addressed in a recent review [2].

Current antiplatelet therapies focus on inhibition of platelet aggregation [19]. However, since the machinery regulating the procoagulant phenotype of platelets is increasingly elucidated, and since the small proportion of procoagulant platelets generated in the growing thrombus appears to play a key role, the procoagulant response should also be considered in the development of new anti-thrombotic strategies [20,21]. For instance, a recent publication by Millington-Burgess and Harper [22] details the peculiar calcium (Ca^2+^) signaling in procoagulant platelets and how this may be targeted for a selective treatment. Similarly, the present review offers a comprehensive overview of the mechanisms underlying procoagulant platelet formation, which may be explored as possible druggable targets.

Platelets are activated by physiological agonists at the site of injury, primarily collagen and thrombin, initiating key specific signaling pathways [23]. They trigger, among others, the rise of cytosolic calcium ([Ca^2+^]_cyt_), secretion of granule content and secondary agonists, and activation of the fibrinogen receptor [24]. Ca^2+^ is at the center of several platelet activation end-points and in particular the procoagulant response, which is induced by a high and sustained [Ca^2+^]_cyt_ [25,26].

This review will first describe the key features of procoagulant platelets focusing on their phenotype, machinery, and functions. Then, we will dissect the intracellular mechanisms underlying the dichotomous activation pattern between aggregating and procoagulant platelets. Finally, we will address the current and future research axes that could be useful to understand the origin of procoagulant platelets and the variability of their response.

## 2. Procoagulant Platelet Characterization

### 2.1. In Search for a Consensus for the Characterization of Procoagulant Platelets

Variations and overlap of procoagulant platelet phenotype are reported [3,27]. In fact, this variation highly depends on the system (adhesion or suspension) and the type of agonists used for platelet activation [3]. Procoagulant platelets have been reported with various names, such as SCIP (sustained Ca^2+^-induced platelets) [28], necrotic 4-(N-(S-glutathionylacetyl)amino) phenylarsonous acid (GSAO) [29], superactivated [30], capped [31], ballooning [32], and zombie [33] platelets. The term “coated” is also used since procoagulant platelets generally express a coating of procoagulant proteins [34]. Here, we employ the term “procoagulant COAT platelets” [1,35,36]. This acronym identifies unambiguously their mode of activation (i.e., induced by COllagen And Thrombin (COAT)) and their coating by retained α-granule proteins [1,35,36,37].

Collagen [38] (and related agonists, such as collagen-related peptide (CRP) [26,39] or convulxin [40,41,42]) and thrombin [43] are the most potent activators [44] of the procoagulant platelet response. Indeed, dual agonist and synergistic signaling mobilize enough Ca^2+^ to form procoagulant platelets while single stimulation by either agonist does trigger poor or moderate procoagulant platelet formation [1,43]. Moreover, since Ca^2+^ influxes are important to trigger the procoagulant response, physiological levels of Ca^2+^ are required in the medium [45,46]. Nowadays, convulxin/thrombin (COAT) dual agonist stimulation in Ca^2+^ buffer—generally recognized as standard—is extensively used by many research groups to assess procoagulant function of platelets [3,4,12,22,47,48,49,50,51]. Common model such as procoagulant COAT platelets are required in order to compare observations between research groups [51]. However, other agonists (e.g., adenosine diphosphate (ADP) [52], arachidonic acid [52], ionophore A23187 [36], or thrombin receptor activating peptide 6 (TRAP6) [52]) or inhibitors (e.g., epinephrine [53] or antibodies [36]) are also used to assess platelet procoagulant response through different pathways.

### 2.2. Delayed Onset of Dichotomous Procoagulant Response

Not all platelets develop procoagulant activity despite receiving the same combined COAT stimulation. In fact, upon activation, they all become aggregant by activating the fibrinogen receptor glycoprotein (GP) IIb-IIIa (also known as integrin αIIbβ3), at first [44,54]. Then, after approximately two minutes [46], cytosolic levels of Ca^2+^ reach a high concentration in some platelets only [26]. This concentration is required to trigger the sequence of events whose terminal end-point is the externalization of negatively charged phospholipids, formation of a coat of proteins, and downregulation of GPIIb-IIIa.

### 2.3. Externalization of Negatively Charged Phospholipids

The main feature of procoagulant COAT platelets is the externalization of negatively charged phospholipids such as phosphatidylserine (PS) and phosphatidylethanolamine (PE) [1,55,56]. Negative phospholipids are the preferential site for adsorption and assembly of the procoagulant factors responsible for the coagulation cascade, namely the intrinsic tenase complex (coagulation factors VIIIa-IXa (FVIIIa-FIXa)) and the prothrombinase complex (FVa-FXa) [57]. The recruitment of the serine proteases of both complexes requires post-translational modifications through vitamin K-dependent gamma-carboxylation in order to facilitate their binding to PS via Ca^2+^ [57]. Presence of PS is usually restricted to the inner side of the platelet wall by the means of inward adenosine triphosphate (ATP)-dependent flippase activity [58]. However, due to the high [Ca^2+^]_cyt_, this flippase activity is inhibited. Moreover, subsequent activation of the transmembrane 16 F (TMEM16F) scrambles PS/PE across the membrane [59,60]. These platelets therefore expose PS, which is one of the prerequisites for the procoagulant activity.

### 2.4. Coating with α-Granule Proadhesive and Procoagulant Proteins

Procoagulant (e.g., factor V/Va) and adhesive (e.g., fibrinogen, fibronectin, VWF, and thrombospondin) proteins secreted from α-granules are retained and irreversibly bound to the surface of procoagulant COAT platelets [34] by a serotonin [36,61] and transglutaminase-dependent mechanism [62]. This specific phenotype has given the alternative name of “coated” platelets [36]. These proteins are involved in the recruitment of other platelets and/or anchoring of the procoagulant platelets in the thrombus. Of note, only procoagulant COAT platelets are coated [27]. Thus coating in addition to PS exposure define procoagulant COAT platelet functionality [1].

### 2.5. Surface Exposure of Cytosolic Factor XIII

Coagulation factor XIII (FXIII), a transglutaminase that stabilizes the fibrin mesh against mechanical stress and proteolysis, is present at high concentrations in platelet cytosol [63]. During convulxin/thrombin activation, platelet FXIII is externalized by a receptor-signaling dependent mechanism on the membrane of procoagulant COAT platelets [64], where it appears to serve several functions, such as cross-linking α2-antiplasmin into fibrin [65], promoting star-like fibrin fiber formation [62], and retaining α-granule proteins [36].

### 2.6. Decreased Aggregatory Properties

Following strong platelet activation, all platelets rapidly activate their fibrinogen receptor and become pro-aggregatory [44,54]. Initially, agonists activate GPIIb-IIIa by an inside-out mechanism [66], increasing its affinity for fibrinogen [66]. The binding of fibrinogen to activated GPIIb-IIIa receptors sustains platelet bridging and mediates also an outside-in signaling culminating in an irreversible aggregation of platelets [66,67,68].

Noteworthy, after a delay of 1–2 min, a fraction of the activated platelets undergoes a phenotypic switch and starts differentiating from aggregating to procoagulant. These platelets are characterized by a progressively downregulated GPIIb-IIIa, which reduces their ability to bind PAC-1, and by the expression of PS [36,46].

GPIIb-IIIa may be inactivated by a calpain-2/TMEM16F mechanism following high and sustained [Ca^2+^]_cyt_ [54]. An alternative hypothesis is that the GPIIb-IIIa receptor might be engaged by a stronger ligand than PAC-1 [61]. According to this model, fibrinogen bound to GPIIb-IIIa is bridged by serotonin and FXIII to other α-granule proteins, such as FV at the surface of procoagulant COAT platelets [61]. Therefore, the strength of binding and the steric hindrance prevent PAC-1 binding and make the GPIIb-IIIa appearing as gradually inactivated.

### 2.7. Cell Membrane Remodeling

After activation, resting discoid platelets [69] flatten and spread extending filopodia that interact with other cells to build the clot [69]. When adhered on collagen, procoagulant platelets undergo lamellipodia formation [70,71] increasing their surface/volume ratio. Subsequently, procoagulant platelets undergo blebbing and ballooning formations [3,21,71]. These are both driven by actin remodeling with contraction of myosin bound to a cortex–membrane linker through a calpain-mediated pathway requiring high [Ca^2+^]_cyt_ to be activated [72]. While blebbing is reversible, ballooning is irreversible because the link with the actin cortex has been disrupted and reformed to stabilize the structure [3]. Ballooning occurs when blebs undergo rapid expansion through water entry through TMEM16F and aquaporin 1 (specific to ballooning) given the high concentrations of ions (sodium (Na^+^), chloride (Cl^-^) or Ca^2+^) entering the cell. Moreover, the hydrostatic pressure maintains the protrusions [3,21,71]. This remodeling is required to increase the surface of procoagulant platelets to bind coagulation factors.

During membrane remodeling, instabilities occur in the membrane forming microvesicles; this event is facilitated under shear stress [23]. Indeed, procoagulant platelets shed microvesicles that also express PS and other procoagulant proteins and, therefore, serve as mobile coagulation sites platforms [73]. The mechanisms underlying microvesicles formation involve high Ca^2+^ concentration, TMEM16F and are also calpain-dependent [74,75] since inhibition of calpain reduces the degree of microvesicles release [75]. The microvesicles membrane composition highly depends on granules exocytosis and fusing after primary activation [76].

### 2.8. Mitochondrial Depolarization

Loss of mitochondrial transmembrane potential occurs during the generation of procoagulant platelets [77]. First, [Ca^2+^]_cyt_ increases after activation through multiple Ca^2+^ channels. Mitochondria accumulate Ca^2+^ during activation [78,79]. Then, the membrane of some mitochondria collapses and becomes depolarized. Therefore, a necrotic spread occurs in cascade, depolarizing the rest of the mitochondria by subsequent Ca^2+^ accumulation. The future procoagulant platelets eventually reach supramaximal [Ca^2+^]_cyt_ required for PS exposure [80].

### 2.9. Sustained “Supramaximal” Cytosolic Ca^2+^ Concentration

Only the platelets that have undergone mitochondria depolarization can reach the so-called “supramaximal” [Ca^2+^]_cyt_ (>100 µM) [26] necessary to trigger the remaining procoagulant end-points. The mitochondrial Ca^2+^ concentration required to trigger the mitochondrial permeability transition pore formation is accumulated through extraction from the internal stores and from the extracellular space with various mechanisms described in Section 3.1.3. Calpain, whose activation relies on this non-physiological Ca^2+^ concentration is a central mediator of all the procoagulant changes [72]. Its activation initiates ballooning (cytoskeletal rearrangement) and PS exposure [17,26].

## 3. Cytosolic Ion Fluxes in Procoagulant COAT Platelets

### 3.1. Ca^2+^ Fluxes

#### 3.1.1. Role of [Ca^2+^]_cyt_ in the Procoagulant Response

Rise of [Ca^2+^]_cyt_ is a marker of platelet activation and mediator of the downstream signaling [26,81]. In procoagulant response expression, Ca^2+^ is a cornerstone of phenotype commitment with the so-called supramaximal Ca^2+^ concentration [26] required to trigger the future sequence of events giving platelets their procoagulant properties. The scramblase TMEM16F as well as calpain have a low sensitivity to Ca^2+^ and are therefore activated by high Ca^2+^ concentrations only [72]. TMEM16F drives PS exposure [22,26] and calpain membrane remodeling [72]. Interestingly, platelet can be forced to express PS by using Ca^2+^ ionophore [1], bypassing all signaling pathways. However, these platelets do not express coating [82] nor downregulate GPIIb-IIIa [1]. This observation highlights a specific mechanism for the generation of procoagulant COAT platelets.

#### 3.1.2. Downregulation

Ca^2+^ homeostasis in platelets is controlled by mitochondria and by three Ca^2+^ channels. Mitochondria internalize Ca^2+^ through the mitochondria Ca^2+^ uniporter (MCU) [78]. The plasma membrane Ca^2+^ ATPase (PMCA) expels Ca^2+^ outside the cell [83]. The sarco-endoplasmic reticulum Ca^2+^ ATPase (SERCA) displaces Ca^2+^ into internal stores [84,85]. These stores are remnants given to platelets from the endoplasmic/sarcoplasmic reticulum (ER/SR) of megakaryocytes [86,87] known as the dense tubular system (DTS) [88]. Additionally, lysosomes-like acidic storages (thought to be lysosomes, dense granules and/or Golgi apparatus [89]) also participate in Ca^2+^ buffering through SERCA3 [88,89]. Finally, the forward mode of sodium–calcium exchangers (NCX) also participates in [Ca^2+^]_cyt_ efflux [90].

#### 3.1.3. Upregulation

Very high and sustained [Ca^2+^]_cyt_ necessary for procoagulant activity are reached through different receptor-operated (ROCE) and store-operated (SOCE) Ca^2+^ entry pathways. Of note, most of the Ca^2+^ influx pathways are considered working in parallel. Understanding the temporality and the hierarchy of these channels might help elucidating their implication in the procoagulant response.

First, agonists activate platelets through their proper receptors. The various signaling pathways activate a core set of intracellular messengers that converge to the activation of phospholipase C (PLC) isoforms. The PLCγ2 isoform is involved in GPVI signaling after collagen activation [91,92]. The PLCβ isoform is downstream of protease-activated receptor (PAR) 1 and 4 activation [24,93]. Then, the PLC isoforms cleave phosphatidylinositol-4,5-bisphosphate (PIP2) into inositol trisphosphate (IP3) and diacylglycerol (DAG) [81]. IP3 interacts with its receptor at the surface of the internal store of Ca^2+^, triggering the release of Ca^2+^ into the cytosol [94,95,96].

Two depletion-independent mechanisms (ROCE) mobilize Ca^2+^ to form the primary wave of Ca^2+^ with the Ca^2+^ released from internal storages. DAG-mediated entry of extracellular Ca^2+^ is operated through the transient receptor potential cation channel subfamily C member 6 (TRPC6) opening [97,98]. TRPC6 is an unselective Ca^2+^-permeable cation channel, which also imports Na^+^ during platelet activation [43]. One pathway of Ca^2+^ entry is also operated through ATP stimulation of P2X1 [99]. The summary of cytosolic Ca^2+^ mobilization regulation is shown in Figure 2.

Depletion of Ca^2+^ from the intracellular store triggers Ca^2+^ mobilization from two main SOCE mechanisms. TRPC1 can be activated either by depletion of intracellular stores [100] or by stimulation of a G_q_-coupled receptor [101]. Stromal interaction molecule 1 (STIM1) is oligomerized when Ca^2+^ stores are depleted triggering ORAI1-mediated Ca^2+^ entry (with cyclophilin A as positive regulator) [96]. TRPC1 and ORAI1 could even be coupled [102].

Finally, an additional Ca^2+^ entry depends on high cytosolic Na^+^ levels. These are responsible for the switching of the NCX functional mode. In its reverse mode, NCX pumps Ca^2+^ in the cytosol while expelling exceeding Na^+^ out of the cell. In addition to NCX activity, Ca^2+^ rise is also dependent on potassium (K^+^)-activated NCX (NCKX) [103].

#### 3.1.4. Oscillating versus Sustained Increase of [Ca^2+^]_cyt_

During the initial Ca^2+^ entry, [Ca^2+^]_cyt_ is referred as oscillatory because platelets can still manage Ca^2+^ efflux (through PMCA, SERCA, and NCX forward mode) and internalization in mitochondria to maintain their homeostasis [104,105]. Afterwards, the NCX reverse mode contributes to achieve the peculiar high and sustained [Ca^2+^]_cyt_ required to depolarize the membrane of mitochondria. Indeed, as the Ca^2+^ concentration rises in the cytosol, the mitochondria act normally as buffer accumulate and store Ca^2+^ through the MCU [78]. Platelets that have stored enough [Ca^2+^]_cyt_ see their mitochondria undergoing mitochondrial permeability transition pores (mPTP) opening [78]. This event releases an important amount of mitochondrial [Ca^2+^]_mito_ in the extracellular space.

### 3.2. Sodium Fluxes

#### 3.2.1. Role of Cytosolic Sodium in the Procoagulant Response

Several research groups reported an increased Na^+^ uptake following thrombin or collagen stimulation [43,44,106,107,108,109,110,111,112]. However, in procoagulant COAT platelets, a sudden Na^+^ increase is followed by a rapid Na^+^ efflux [44]. The sudden Na^+^ mobilization after strong activation controls the reversing of the NCX mode, which is critical for procoagulant COAT platelet formation [42]. Na^+^ entry is also associated with concomitant water entry [113]. This platelet swelling drives PS exposure by enhancing exocytosis through an unknown mechanism [113].

#### 3.2.2. Downregulation

Na^+^ efflux is reported to be mediated through Na^+^/K^+^ exchanger [113] and by the NCX reverse mode [42,114].

#### 3.2.3. Upregulation

Na^+^ enters platelet cytoplasm via the NCX forward mode [42,44], Na^+^/H^+^ exchanger [115,116], non-selective channels, such as ATP-receptor P2X1 [117], glutamate-receptor AMPA [118], and epithelial Na^+^ channel (ENaC) [109], or DAG-mediated TRPC3/6 [43,113,119]. Increasing intracellular concentration of Na^+^ with ouabain (blocking Na^+^ exit via Na^+^/K^+^ exchanger), gramicidin (Na^+^ ionophore), or monensin (simulating the action of Na^+^/H^+^ exchanger) induces PS exposure [27,113,120].

### 3.3. Chloride Fluxes

Several families of Cl^-^ channels are found in platelets, such as chloride channel proteins (CLCN), chloride intracellular channels (CLIC), and calcium-activated chloride channels (CaCC) within the TMEM16 family (such as TMEM16A, TMEM16B, and TMEM16F) [121]. Cl^-^ is required for full platelet function. Murine platelets from a CLIC1 knockout model present decreased ADP-induced platelet activation [122]. Thrombin activation of human platelets in Cl^-^-free buffer resulted in reduced aggregation [123]. Procoagulant ballooned platelet formation was reduced in Cl^-^ or Na^+^-free medium [71]. Cytosolic Cl^-^ increases during the development of procoagulant activity [71]. Cl^-^ channel inhibitors reduce PS exposure on the platelet surface after COAT activation [124]. Of note, [Ca^2+^]_cyt_ mobilization was also impaired in presence of Cl^-^ channel inhibitors or with platelet activation performed in Cl^-^-free buffer. However, inhibitors of Cl^-^ channels did not affect PS exposure mediated by Ca^2+^ ionophore. Since Cl^-^ blockers (or absence of Cl^-^ in the buffer) only affect agonist-dependent PS exposure and not ionophore-induced PS-exposure, Cl^-^ entry is expected to play a role in the signaling—upstream of TMEM16F—to achieve a full procoagulant activity [124]. The exact Cl^-^ signaling is not fully elucidated but it appears that platelet activation and early Ca^2+^ mobilization induce an initial Cl^-^ entry inter alia via CaCC (other than TEMEM16F, which requires higher Ca^2+^ levels than TMEM16A/B to be activated [125]). On the other hand, if Cl^-^ would rather play a role of counter-ion, its sudden increase in response to Ca^2+^ entry would help to stabilize plasma membrane hyperpolarization. Indeed, hyperpolarization becomes an important driving force for further Ca^2+^ entry [124].

### 3.4. Potassium Fluxes

Platelets experience K^+^ leaking during their activation process [44,126,127]. Cytosolic K^+^ of platelets is regulated through voltage-gated K^+^ channels (Kv1.3) [128] and by three types of Ca^2+^-activated K^+^ channels [129,130]: large-conductance (BK) channels, intermediate conductance (IK) channels, and small-conductance (SK) channels. Elevated and persistent Ca^2+^ levels have been linked to K^+^ leaking through Ca^2+^-sensitive K^+^ Gardos channels [130,131]. The latter (IK channels) are half-maximally activated with 950 nM Ca^2+^ concentration [132], whereas SK and BK channels are half-maximally activated with 400–800 nM [131] and 4 µM [133], respectively. Accordingly, Gardos channel deficiency in mice did not suppress the PS exposure and procoagulant response [134]. Efflux of K^+^ in platelets was reported during classical activation and procoagulant phenotype induced by COAT [42,44,129,130]. In light of its regulation and implication in procoagulant response, K^+^ seems to play the role of positive charge exchanger for the massive Ca^2+^ entry.

## 4. Mechanisms Involved in Procoagulant COAT Platelet Formation

### 4.1. Surface Receptors

#### 4.1.1. GPVI

Collagen participates in two ways to platelet activation. Collagen immobilizes platelets via their adhesion to GPIa-IIa (integrin α2β1) and initiates an important intracellular signaling through GPVI, which is crucial for procoagulant response [70,135]. GPVI is a transmembrane protein, which forms a complex with the Fc receptor γ-chain (FcRγ). Indeed, once activated by the binding of collagen, GPVI recruits Src-family kinases—Src, Lyn, and Fyn—to phosphorylate the two conserved tyrosines in the immune-receptor tyrosine activation motif (ITAM) of the FcRγ chain [136,137]. This leads to the binding and activation of Syk to phosphorylate the linker for activation of T cells (LAT), which initiates the formation of a signalosome trough the recruitment of Grb2, Gads, SLP-76, and PI3K [138,139]. This complex leads to Bruton’s kinases-mediated phosphorylation of PLCγ2 [91,92]. PLCγ2 cleaves phosphatidylinositol-4,5-bisphosphate (PIP2) into IP3 and DAG [81]. IP3 initiates the [Ca^2+^]_cyt_ mobilization and Ca^2+^/DAG-mediated guanine exchange factor I (CalDAG-GEFI) signaling, while DAG activates protein kinase C (PKC). Both PKC and CalDAG-GEFI activate the small-GTPase Rap1 signaling, resulting in integrin activation and secretion of granule and soluble agonists [138,139,140]. The signaling downstream of GPVI and its involvement in the procoagulant response are summarized in Figure 3.

Although collagen alone seems to induce more PS exposure on the platelet surface than thrombin alone [141,142], single agonist activation does not achieve a very high expression of PS [1,23,70]. Platelets become highly procoagulant either when activated with collagen in the presence of shear stress [23] or with simultaneous activation by collagen and thrombin [1]. In the latter situation, the peculiar signaling of procoagulant platelets is orchestrated through the simultaneous activation of the collagen and thrombin receptors [27,143].

#### 4.1.2. Protease Activated Receptors (PAR) 1 and 4

After vascular injury, tissue factor (TF) is expressed by damaged endothelium [144]. A low thrombin signal is generated by the initiation phase of the coagulation cascade driven by TF/VIIa. Thrombin serves as a primary activator of platelets [144]. Thrombin binds PAR 1 and 4 and cleaves their N-termini, uncovering a tethered ligand to activate the receptor itself [145]. PAR1/4 are members of the G protein-coupled receptor (GPCR) superfamily [146]. Activated PAR1 and PAR4 signal through G_q_ and G_12/13_ pathways. G_q_ activates PLCβ to form IP3 and DAG, inducing [Ca^2+^]_cyt_ rise and CalDAG-GEFI signaling, and PKC activation, respectively. G_12/13_ trigger the reorganization of the actin cytoskeleton (shape change) mediated by RhoA pathway [24,93].

PAR1 works already at low thrombin concentrations and is the principal receptor for thrombin, while PAR4 needs higher thrombin concentrations to be activated [143]. PAR1 is therefore also central for the procoagulant signaling of platelets as its inhibition reduces COAT induced generation of procoagulant platelets [45], and PAR4 contributes the least to PS exposure [147,148]. However, agonists for PAR1 or PAR4 or a combination of them induce a weaker Ca^2+^ mobilization and PS exposure than thrombin does, and these enhance less efficiently collagen co-activation [44]. This difference may reflect the additional interaction of thrombin with the GPIb receptor [149]. Ionophore A23187 (triggering PS exposure but not coating [82]), in addition to thrombin stimulation, revealed a thrombin-induced implication of FV binding [36]. The signaling downstream of PAR1/4 and GPIb-IX-V stimulation is summarized in Figure 4.

#### 4.1.3. GPIb-IX-V

The GPIbα component of the GPIb-IX-V complex binds VWF and other ligands, such as thrombin and coagulation factor XI [151]. While thrombin stimulation seems to require the sole presence of GPIb [149,152], GPIb-IX has been reported as necessary in VWF-mediated stimulation [150].

Thrombin at low doses binds GPIb, which is also referred to as “low density high affinity” thrombin receptor [149,152], facilitating its interaction with PAR1 as well [153]. The thrombin-dependent GPIb signaling engages the protein 14–3-3ζ, which activates Src family kinase (SFK). The latter is also mediated by PAR1/4 stimulation at low thrombin levels. SFK mediates Rac1, which controls phosphoinositide 3-kinase (PI3K) activity. At high thrombin concentrations [153], PAR4 signaling, in addition to the PAR1 and GPIb, intervenes to obtain the full thrombin-induced response [149,152] as shown in Figure 4.

GPIb in addition to PAR signaling seems to enhance the thrombin component in the context of COAT stimulation [154]. In fact, although blocking GPIb did not affect Ca^2+^ mobilization and procoagulant activity [45,154], a minor decrease in PS exposure and thrombin generation induced by COAT platelets were reported in platelets from Bernard–Soulier syndrome patients [154,155,156]. Indeed, the signaling produced by GPIb-IX-V is redundant with GPVI signaling with regard to the syk pathway. This may explain the limited implication of the GPIb-IX-V complex in the procoagulant response of COAT platelets [149,150,157]. Of note, in late activation state, the GPIb-IX-V complex becomes a site of procoagulant factors adhesion [158].

#### 4.1.4. P2Y12

ADP is a secondary platelet activator leading to a wide range of responses. For instance, activation of the ADP receptor P2Y12 amplifies the procoagulant response elicited by collagen and thrombin by two mechanisms. This amplification is mediated, firstly, by the activation of PI3K controlling granules secretion. The second element of the amplification is the sensitization to the activation [159,160]. This phenomenon results in a decrease in cyclic adenosine monophosphate (cAMP), which increases [Ca^2+^]_cyt_ [143,161,162,163,164]. cAMP activates protein kinase A (PKA), which is an inhibitor of the IP3 receptor, thus reducing Ca^2+^ mobilization after activation [165]. This is in line with the reduced potential to generate procoagulant platelets when ADP receptors P2Y12 are desensitized [49] or inhibited (e.g., by clopidogrel or cangrelor) [161,162].

#### 4.1.5. Receptors That Are Regarded as Less Relevant for the Procoagulant Response

In platelets, activation of P2Y1 by ADP initiates the increase in intracellular Ca^2+^ by PLCβ-mediated cleavage of PIP2 in IP3 (Ca^2+^ mobilization) and DAG (shape change and granule release). This mechanism is redundant with thrombin signaling and might explain the low relevance of the P2Y1 receptor in the context of COAT platelets [159,166].

The ATP-gated ionotropic receptor channel P2X1 is a non-selective cation channel. Its activation triggers direct Na^+^ and Ca^2+^ entry [22,99,160]. However, P2X1 has a positive implication on PS exposure only under low collagen concentration with a moderate shear stress or low to moderate thrombin concentration [167]. Although P2X1 is irrelevant for Ca^2+^ mobilization after platelet COAT activation, its Na^+^ influx activity may contribute to the reversing of the NCX mode.

Thromboxane A2 (TXA2) is produced de novo and secreted following platelet activation. The activated form of the phospholipase A2 (PLA2) releases arachidonate from membrane phospholipids [168]. The latter is transformed by the cyclooxygenase (COX-1) and the thromboxane synthase in TXA2. The activated TXA2 receptor, called TP, triggers the same intracellular mechanisms [169,170] as the ADP receptor P2Y1, thus sustaining shape change, degranulation, and aggregation. Even though TXA2 increases Ca^2+^ mobilization induced by thrombin [44], it does not amplify the procoagulant response in COAT, as also reported with its analogue U46619 [163] or as implied by the poor impact of aspirin treatment on the generation of procoagulant platelets [50,171].

### 4.2. Key Regulators of Intracellular Signaling

#### 4.2.1. PLC/PKC Isoforms

The activated PLC isoforms (β and γ2, both involved in COAT platelets) trigger the formation of IP3 and DAG, which mediates PKC activation [24]. PKC isoforms are involved in the regulation of diverse platelet activation processes. Of note, Ca^2+^ differently regulates PKC isoforms: “classical” PKC isoforms α and β contain a Ca^2+^-binding domain, while more recently described “novel” isoforms, such as δ or θ, lack these Ca^2+^-binding domains and are therefore not primarily implicated in Ca^2+^-mediated processes [172].

Each of these PKC isoforms has its peculiar role and agonist-specific regulation [172,173]. One described role for PKCα is to increase Ca^2+^ mobilization and procoagulant activity [46,174]. The phosphorylation site T497 of PKCα is upregulated during COAT stimulation [46]. The function of this phosphorylation is however not understood yet. Moreover, PKCα enhances Ca^2+^ mobilization in the cytosol by triggering the reverse mode of NCX [174]. On the other hand, PKCβ is an amplifier of reactive oxygen species (ROS) production, triggering mPTP opening and, therefore, PS exposure [175].

In thrombin-induced activation, classical PKC isoforms are activated and, in addition to increased SERCA activity, an increasing Na^+^/K^+^ exchanger activity but with a reduced NCX forward mode activity is observed [176]. On the other hand, in collagen-derived activity, classical PKC isoforms increase intracellular Ca^2+^ mobilization, whereas novel isoforms (δ or θ) reduce its mobilization [173]. The differences between thrombin and collagen-derived PKCα/β activity point to an unknown mechanism of interest in COAT platelets. One hypothesis is that phosphorylation of PKC isoforms occurs at specific sites and time points as a function of the activator [177]. For instance, after GPVI and GPIb-IX-V engagement, the preferential and early activation of PCKα is mediated by its proximity to Src and Syk [178,179] and modulated by a feedback loop inhibiting Syk [180].

The novel isoform PKCθ reduces Ca^2+^ signaling from GPVI activation, which results in impaired PS exposure [173,181] and its absence demonstrated to increase PS exposure [182]. Studies have shown that PKCθ is involved in GPIIb-IIIa outside-in signaling [183]. Taken together, these observations may explain why platelets from Glanzmann thrombasthenia patients lacking GPIIb-IIIa show higher procoagulant response [37]. Of note, PKCθ deficiency leads also to impaired GPIIb-IIIa engagement and a reduced thrombin-induced aggregation [184,185]. PKCθ might in fact be involved both in inside-out and in outside-in signaling via coupling and uncoupling of PKCθ with the GPIIb-IIIa receptor [186]. Regarding PKCδ, the current knowledge does not seem to indicate any modulation on the procoagulant activity [172]. This isoform seems rather linked to exocytosis/secretions events.

#### 4.2.2. p38 Mitogen-Activated Protein Kinases (p38 MAPK)

p38 MAPK are serine/threonine kinases that regulate cellular processes, such as migration, differentiation, survival, and apoptosis [187,188]. p38 MAPK are phosphorylated downstream of different physiological platelet agonists; however, some uncertainty remains about the specific function of p38 MAPK in the platelet activation process [187]. During the procoagulant response, significantly lower p38 MAPK phosphorylated states were observed in procoagulant platelets compared to aggregating non-procoagulant platelets [46]. However, other studies reported that p38 MAPK are not implicated in the rapid procoagulant PS exposure [70,189], while their inhibition prevented platelet apoptosis [189]. Given the debate existing about the procoagulant activity of apoptotic platelets, further studies must be carried out about the implication of the p38 MAPK [187].

#### 4.2.3. NCX

A sudden Na^+^ mobilization after strong activation in COAT platelets precedes the reversing of the NCX mode [42]. Our group demonstrated that reversal of NCX occurs only in the procoagulant fraction of platelets, leading to an additional Ca^2+^ influx, while non-procoagulant (aggregant) platelets keep NCX in the forward mode (Ca^2+^ efflux), despite a similar initial strong activation [42]. The forward mode of NCX 1 and 3 sustains [Ca^2+^]_cyt_ efflux [90]. Ca^2+^ is expelled from the cell using the electrochemical gradient of Na^+^ with a ratio of 3Na^+^:1Ca^2+^ [190]. However, the NCX forward mode can be inverted to increase [Ca^2+^]_cyt_. The reverse mode pumping Ca^2+^ in the cytosol while expelling exceeding Na^+^ promotes further Ca^2+^ entry and thus procoagulant signaling [43,191,192].

Scarce data exist on regulation of the NCX reverse mode in COAT platelets. The reversing of the NCX mode is regulated by PKCα activation [174] after COAT stimulation [46]. The NCX reverse mode is also regulated by Na^+^ entry through TRPC6 [43].

#### 4.2.4. Mitochondria

Mitochondria accumulate Ca^2+^ through the MCU complex [78], while [Ca^2+^]_mito_ efflux is mediated by mitochondrial H^+^/Ca^2+^ and Na^+^/Ca^2+^ exchangers [193]. Inactivation of the MCU impairs PS exposure (among other activation end-points), thus highlighting the involvement of mitochondria in the procoagulant response [78]. In fact, high and sustained [Ca^2+^]_cyt_ required for PS exposure to occur are reached only after mPTP opening, releasing an important amount of [Ca^2+^]_mito_ into the cytosolic space [26]. Indeed, inhibitors of mPTP opening reduce the formation of procoagulant platelets, while activators of mPTP enhance their production [194]. mPTP opening is dependent on a critical [Ca^2+^]_mito_ threshold depending on cyclophilin D (CypD) [78,79] and is facilitated in presence of ROS [195,196]. Genetic deletion of CypD or its inhibition with cyclosporine A resulted in impaired opening of mPTP and PS exposure as well as microparticles formation [80]. ROS generated in activated platelets such as H_2_O_2_ can also trigger mPTP opening in a CypD-dependent manner [196]. Moreover, mPTP opening is required for calpain-mediated GPIIb-IIIa inactivation as persistent GPIIb-IIIa activation was observed both in CypD^-/-^ or MCU^-/-^ mice [46,78,197].

#### 4.2.5. Scramblase

PS exposure through scramblase activity is the terminal end-point of the procoagulant response. ATP-independent but Ca^2+^-dependent scramblase activity of the TMEM16F (also known as Anoctamin 6 (Ano6)) is responsible for PS exposure in platelets. TMEM16F equilibrates the PS/PE (normally restricted to the inner cell membrane) across the membrane and therefore confers a procoagulant state to the cell [59,60]. PE grants even a greater procoagulant activity than PS [198]. TMEM16F has a low affinity for Ca^2+^, which explains why its activity is shown only at a [Ca^2+^]_cyt_ of 1–10μM [199] solely obtained in procoagulant platelets [22,26].

### 4.3. Extracellular Outside-In Signaling of GPIIb-IIIa in Aggregation

All platelets activate their GPIIb-IIIa receptors through Ca^2+^ mobilization and key activators such as PI3K and PLC isoforms [66,93]. Irreversible aggregation seems to be mediated by PKCθ [183,200]. In procoagulant platelets, GPIIb-IIIa is inactivated by a calpain-dependent mechanism when supramaximal Ca^2+^ concentrations are reached [54]. However, whether the outside-in signaling also participates in procoagulant platelet formation is not elucidated.

Fibrinogen, fibrin, or VWF bind to the activated form of GPIIb-IIIa, which triggers the so-called outside-in signaling [201] by the resultant conformational change. This initiates irreversible aggregation, clot retraction, release of molecules, and spreading of platelets [200]. Briefly, clustering of the activated and bound form of the GPIIb-IIIa induces the activation by autophosphorylation of the Src factor. Src activates by phosphorylation a wide range of factors such as the focal adhesion kinase, Syk kinase, GTPase-activating proteins mediating Rho factor (RhoGAP), Rac-GEFs, RhoGEFs, and PI3K. Gα13, talin, kindlin, tensin, and vinculin make the links between the integrin β3 cytoplasmic tail and actin. This interconnected actin net triggers and synchronizes the cell shrinking and clot contraction.

The early response of GPIIb-IIIa outside-in signaling involves Src-family kinases and Syk [201] as in GPVI signaling. However, in patients suffering from Glanzmann thrombastenia, the procoagulant potential tends to be higher than reference values [37]. Rosado et al. [202] indeed showed that outside-in signaling blocks further Ca^2+^ mobilization and cytoskeleton rearrangement necessary for procoagulant response. Therefore, GPIIb-IIIa outside-in signaling has so far not shown any positive implication in procoagulant response and could be one of its negative regulator [37].

## 5. Potential Drivers of Platelet Phenotypic Diversification

It would be tempting to hypothesize the existence of a single mechanism fine-tuning the procoagulant platelet response. However, in light of the person-to-person variability observed in the ability to generate procoagulant COAT platelets [2] and the complexity of the underlying signaling pathways, it is unlikely that only one mechanism controls such variation. The next sections will summarize potential intrinsic and extrinsic drivers of the procoagulant diversification as shown in Figure 5. Furthermore, the impact of the in vivo environment on the generation of procoagulant platelets will be discussed.

### 5.1. Intrinsic Platelet Heterogeneity

When delivered into the blood stream, platelets have been attributed a variable set of organelles and granules with a variable content [203,204,205]. This influences their size [205] and density [206,207], and by extension their biochemical reactions and ultimately the platelet response. Podoplelova et al. [5] have indeed proposed the combination of platelet size, age, and number of mitochondria as factors determining the procoagulant response.

#### 5.1.1. Platelet Size

Handtke et al. [208,209,210] extensively studied and reviewed size-based platelet subpopulations and their intrinsic and functional differences. Platelet size does not seem to correlate with platelet age in steady state thrombopoiesis [211]. Large platelets express up to 50% more GP and among them GPIb, GPIIIa, GPVI, and P2Y12 [210]. Moreover, they adhere faster and cover a larger area on collagen [210]. They also exhibit higher reactivity (shorter lag time) to ADP and collagen [210]. Of interest, large platelets are more likely to expose PS after dual TRAP6/convulxin stimulation [208,210]. Large platelets are often associated with a pro-thrombotic phenotype [212,213]. Larger platelets undergo higher protein phosphorylation [210].

#### 5.1.2. Number and Contents of Granules

Some studies evoke the fact that lower density platelets exhibit higher sensitivity to thrombin with higher Ca^2+^ response and lower level of the phosphorylated (functional) form of the vasodilator-stimulated phosphoprotein [214]. Low density platelets also show lower inhibitory response to nitric oxide and smaller increase in cAMP to prostaglandin E1 addition than in high density platelets [207,214]. This could be due to higher α-granules content although it is controversial [215]. Moreover, secretion is performed granule by granule or with granules cluster according to the strength of stimulation [216]. In the context of the dichotomous procoagulant response, the heterogeneity of granules content and secretion is, to the best of our knowledge, not well known [217].

#### 5.1.3. Number of Mitochondria

Wiskott–Aldrich syndrome, caused by mutations in the Wiskott–Aldrich syndrome protein (WASP) regulating actin activity, is characterized by increased clearance of platelets due to early PS exposure and lower mitochondrial count [218]. In line with this observation, the work of Obydennyi et al. [218] shows that platelets with less than five mitochondria are more prone to express PS than platelets with a higher number. This could be explained by a lower buffering power of excess [Ca^2+^]_cyt_ in the presence of a low number of mitochondria. Therefore, mPTP opening as well as PS exposure would occur more rapidly in those cells.

#### 5.1.4. Receptor Density and Reactivity

Interestingly, Sveshnikova et al. [105] have proposed and simulated in silico [219] that the high and sustained [Ca^2+^]_cyt_ required for PS exposure might be due to a two-step “decision-making” mechanism. First, the low (ca. 1000 per platelet) and highly variable surface density of the PAR1 receptor between individuals and allegedly between platelets correlates with the platelet response [220]. Similarly, variations in surface GPVI between individuals modulate platelet procoagulant activity [221]. Second, an heterogeneity in the number of Ca^2+^-pumps may explain the variability of [Ca^2+^]_cyt_ achieved after a similar stimulus and the fact that only a fraction of platelets becomes procoagulant [105]. Another source of variability may be the NCX, whose reverse mode of function adds another Ca^2+^ entry pathway, regulated by potentially variable local [Na^+^]_cyt_ [42]. These studies show that inter-platelet heterogeneity in surface receptor density should be deepened in order to understand the variation in the procoagulant response formation.

### 5.2. Age-Dependent Alterations

#### 5.2.1. Apoptotic Platelets Exhibiting PS Exposure Are Not Functionally Procoagulant

Aging of the platelets is an intrinsic heterogeneity factor. Aged platelets expose PS through an apoptotic-like mechanism [222,223]. However, platelets appear to become functionally procoagulant only after a necrotic, agonist-induced mechanism [4,29,222,223]. Indeed, since apoptotic PS-exposing platelets are normally cleared by autophagy before being involved in a procoagulant response, it is unlikely that old cells can support a procoagulant function [222,224].

#### 5.2.2. ROS Accumulated during Aging and Their Role

Platelet aging induces ROS accumulation within the cell [225]. In accordance, Agbani et al. [3] hypothesized, based on [226,227], that ROS in platelet aging might predetermine platelets to become procoagulant. ROS generated during activation are indeed an important driver of the procoagulant response [228]. Such non-causative but associative mechanism might reinforce the idea that a platelet might need a certain package that makes it “decide” to become procoagulant. In the other hand, younger platelets, as monitored with the mRNA dye thiazole orange, appear to have a greater procoagulant potential, detected, e.g., by α-granule FV/Va bound on their surface [1]. The role of intracellular ROS prior platelet activation therefore remains to be determined.

#### 5.2.3. Shedding of Surface Receptors with Aging and Strong Activation

Shedding of platelet receptor ecto-domains occurs with aging, both in vitro (storage lesion of platelet concentrates) and in vivo. The A disintegrin and metalloprotease 17 (ADAM17) constitutively cleaves GPIbα that becomes less available for the adhesion with in vivo aging [229,230]. GPVI also undergoes alteration during storage [49]. Of note, in a canine model GPVI function declines progressively as platelets age in vivo as well [231]. These receptors are therefore less responsive and lead to decreased platelet aggregation and PS exposure [232,233]. Therefore, receptor modifications due to aging could be a determining factor decreasing the procoagulant response.

Acute GPIbα and GPVI shedding also occurs upon strong activation [234]. GPVI is cleaved upon activation by ADAM10 [229]. Shedding of surface receptors coincides with PS exposure and is mediated by a pathway of potential interest in COAT platelets. This pathway depends on elevated Ca^2+^ induced by ionomycin and convulxin/thrombin [234]. Dichotomous platelet receptor shedding could confer to the terminally activated platelets a specific functionality. For instance, GPIbα shedding reduces VWF binding, whereas procoagulant factor binding is enhanced, as observed by Baaten et al. [234].

### 5.3. Variability of Activation Pathways

#### 5.3.1. NCX Reverse Mode

As depicted in Figure 4, the NCX reverse mode has a crucial implication in the procoagulant response [42]. In fact, the NCX reverse mode supports the additional source of Ca^2+^ entry thought to be necessary for mitochondria overload and mPTP formation eventually triggering PS exposure.

#### 5.3.2. Protein Phosphorylation

Upregulation of the phosphorylation at T497 of the PKCα has been observed in procoagulant compared to aggregant platelets during COAT stimulation [46]. On the other hand, downregulation of the phosphorylation at the sites T180 and Y182 of p38-MAPK protein has been found in procoagulant compared to aggregant platelets [46]. The phosphorylation status of CypD can modulate the threshold of [Ca^2+^]_mito_ required to open mPTP [235,236]. Variation in the phosphorylated form of CypD can decrease the Ca^2+^ concentration required to open mPTP or even render mPTP insensitive to calcium accumulation.

#### 5.3.3. GPIIb-IIIa Outside-In Signaling

As proposed by Mischnik et al. [68], upon outside-in signaling, Src and tyrosine phosphatases demonstrate an interplay that creates a bistable switch between reversible and irreversible aggregation. The extent of such a phenomenon is currently unknown. However, it is tempting to hypothesize that only platelets not undergoing stable and irreversible aggregation are able to express procoagulant response. According to Topalov et al. [237], GPIIb-IIIa outside-in signaling also seems to be involved in a subpopulation of PS-exposing but low [Ca^2+^]_cyt_ platelets, which still retains some pro-aggregatory properties and mitochondrial potential. They represented a maximum 15% of the total platelet population after thrombin activation. The functional procoagulant response of this population, which does not fit the consensus definition of COAT platelets, has not yet been studied.

### 5.4. Rheology and Cell–Cell Interactions

In vitro experiments in suspension do not reflect the complexity of in vivo platelet activation with the whole spectrum of physiological agonists present at various concentrations in the growing thrombus. Indeed, clot architecture drives agonist concentrations, cell-to-cell interactions and therefore platelet responses. Intra-vital experiments with mice can represent more accurately how the phenotypical diversification occurs among the platelets.

#### 5.4.1. Local Agonist Concentration

As presented below in Section 5.4.3., the change in porosity within the clot contributes to the accumulation of agonists in the core of the thrombus. Since platelets are sensitive to paracrine signals as well as their own through granules secretion, heterogeneous response can arise. For example, Bandyopadhyay et al. [238] have proposed that ADP-dependent aggregation lead to either stable aggregation or disaggregation in a dose-dependent manner. Similar behaviors might be observed with other agonists. For instance, thrombin at low doses requires GPIb-IX-V, serving as a “low-density high-affinity” receptor for interacting as for PAR1 as well [44,149,152].

Within a growing thrombus, close vicinity of a platelet to another might help to synchronize their behavior [32]. After both outside-in signaling of aggregated platelet and local thrombin generation by the procoagulant subpopulation, the clot retracts, and extracellular Ca^2+^ accumulates within the thrombus and triggers a synchronous ballooning and micro-vesiculation of the platelets [32].

#### 5.4.2. Shear Stress

Procoagulant platelets can be elicited by adherent collagen [23,52] or thrombin [23] under shear stress. Up to 90% of the platelets become procoagulant platelets when shear stress (Rac1-dependent) is added at 6000s^−1^ [23]. However, shear alone does not induce a procoagulant response in absence of collagen [23]. Shear affects VWF conformation, which depends on the force-induced stretching of the molecule. The affinity of GPIb for VWF increases with greater shear-stress [239]. As washed platelets (without plasma VWF) were used in the studies addressing shear-induced response [23,52], the extent of VWF conformation changes in the procoagulant response is not known.

#### 5.4.3. Architecture of the Clot

Gain of procoagulant activity at the expense of aggregatory properties leads to a heterogeneous structure of the thrombus [18,41]. To figure out how subpopulations could be arranged in the thrombus is a new challenge. As depicted in Figure 1, it is thought that the first platelets interacting with exposed collagen are fully adherent and flat, in order to cover as much as possible the lesion [88]. Whether these platelets eventually become procoagulant COAT platelets, promoting generation of thrombin and its diffusion to the core of the thrombus remains speculative. In fact, platelet procoagulant differentiation within the thrombus in a spatio-temporal way has been scarcely studied.

Brass et al. [16,240,241] have made very interesting observations about agonists and activated platelets distribution within the hemostatic plug in vivo. According to their work, a clot is composed of a tight core of aggregating platelets, which creates a region of reduced transport that facilitates local thrombin accumulation and greater platelet activation. The shell of the thrombus is composed of loosely adherent/aggregated cells. Due to the higher porosity [16], platelets are activated by a gradient of soluble agonists [240] such as ADP and TXA2 that are washed out by the flow. The platelets composing the shell express lower levels of P-selectin. The presence of highly activated platelets, namely, procoagulant platelets, has been observed in the core of the thrombus by Hayashi et al. [104]).

Munnix et al. [41], observed the formation of a single string of procoagulant platelets at the periphery of the thrombus. In accordance with the model shown in Figure 1, shedding of GPIb and GPVI [234] and GPIIb-IIIa downregulation [36,54] redirect the functionality of procoagulant platelets. It has indeed been proposed that non-aggregating procoagulant platelets are mechanically translocated (“squeezed”) from the core to the outer shell of the clot by irreversibly aggregated platelets mediating clot contraction [18]. At the surface of the thrombus, procoagulant platelets could therefore continue to consolidate the clot by fibrin deposition and, at the same time, limit its expansion because no other platelets could be recruited by VWF through the missing GPIb-IX-V complex [18,29] and/or aggregated by the downregulated GPIIb-IIIa [197,242].

## 6. Conclusions

Procoagulant COAT platelets, formed upon combined activation by collagen and thrombin at sites of vascular injury, localize and enhance the generation of thrombin to activate other platelets and deposit fibrin in order to consolidate the clot. In addition, by losing aggregatory properties, they limit clot expansion. Platelets respond to injury within seconds, start procoagulant diversification after 1–2 min, and require several minutes to build a stable clot. Within this time frame, platelets cannot replicate or perform extensive protein synthesis. Therefore, rapid mechanisms such as ion fluxes and post-translational modifications of signaling proteins regulate their response.

Future procoagulant COAT platelets need to accumulate enough [Ca^2+^]_cyt_ to trigger specific key events such as mitochondria depolarization. The ensuing high and sustained [Ca^2+^]_cyt_ triggers the functional response composed of PS exposure, formation of the tenase and prothrombinase complexes, coating with adhesive factors and FXIII, GPIIb-IIIa downregulation, and GPIb and GPVI shedding.

While recent research has revealed the key role of cytosolic Na^+^ and in particular of the reverse mode of NCX for the formation of procoagulant platelets, the regulatory upstream mechanisms are not yet elucidated. Intrinsic platelet heterogeneity and extrinsic factors may explain the dichotomous procoagulant response and the high variability observed between individuals. Studies investigating the generation of procoagulant COAT platelets in three-dimensional models will contribute deciphering their role in vivo.

## Figures and Tables

**Figure 1 ijms-23-02536-f001:**
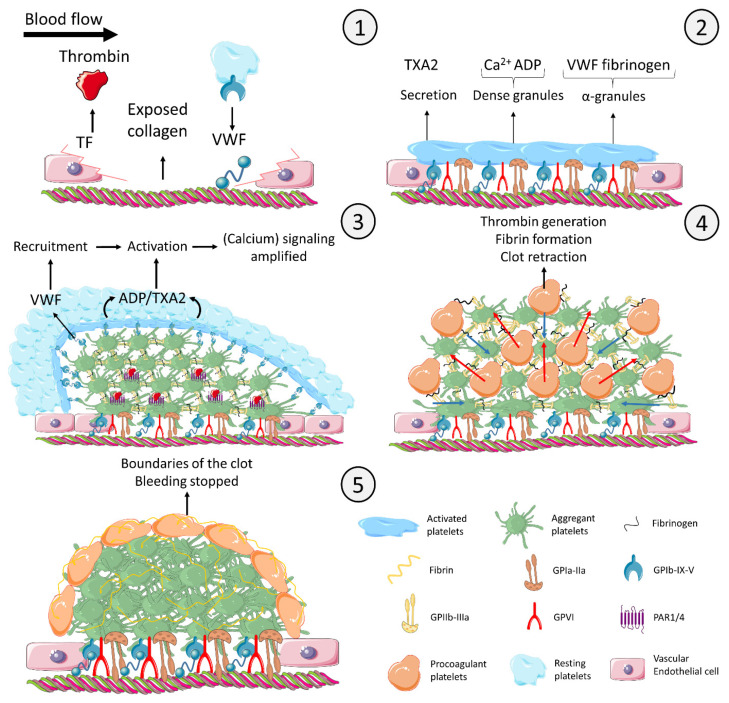
Model of time-and agonist-dependent formation of the thrombus and diversification of platelets at the site of injury. **Panel 1. Initiation**: The injured endothelial cells express TF and secrete VWF. The subendothelial collagen becomes exposed. Initial thrombin is produced due to TF expression. Platelet GPIb-IX-V complex interacts with the VWF deposited on collagen. **Panel 2. Secretion and shape change**: Platelets bind to collagen via the receptor GPIa-IIa. Even though GPIb-IX-V and GPIa-IIa trigger intracellular signaling, platelet activation is mainly mediated by the collagen receptor GPVI. Activated platelets synthesize and secrete TXA2. Dense granules containing Ca^2+^ and ADP and α-granules containing VWF and fibrinogen are secreted. **Panel 3. Aggregation and endoluminal thrombus growth**: Platelets aggregate by means of fibrinogen bridging activated GPIIb-IIIa receptors (integrin αIIbβ3). Secreted TXA2 and ADP activate platelets that are recruited into the growing thrombus by VWF secreted from α-granules. Initial thrombin generation in combination with collagen further activates platelets in the thrombus core. **Panel 4. Procoagulant platelets**: As the thrombus is growing and the platelets become more and more activated, the increasing cytosolic level of calcium eventually creates procoagulant platelets. Procoagulant platelets downregulate their GPIIb-IIIa receptor, are coated with fibrinogen and other adhesive proteins to be retained in the thrombus, and express negatively charged phospholipids binding coagulation complexes, which localize and enhance thrombin generation. Thrombin converts fibrinogen into fibrin, which consolidates the platelet clot. In parallel, the outside-in signaling of aggregating platelets inducing clot retraction (blue arrows) squeezes procoagulant platelets out of core of the thrombus (red arrows). **Panel 5. Termination**: The final configuration completely stops the bleeding and defines the boundaries of the clot to avoid any undesired expansion. The figure uses modified images from Servier Medical Art under a Creative Commons Attribution 3.0 Unported License (http://smart.servier.com, accessed on 31 January 2021). This figure is inspired from [14,15,16,17,18]. Legend: ADP: adenosine diphosphate; Ca^2+^: calcium; GP: glycoprotein; PAR: protease activated receptor; TF: tissue factor; TXA2: thromboxane A2; VWF: von Willebrand factor.

**Figure 2 ijms-23-02536-f002:**
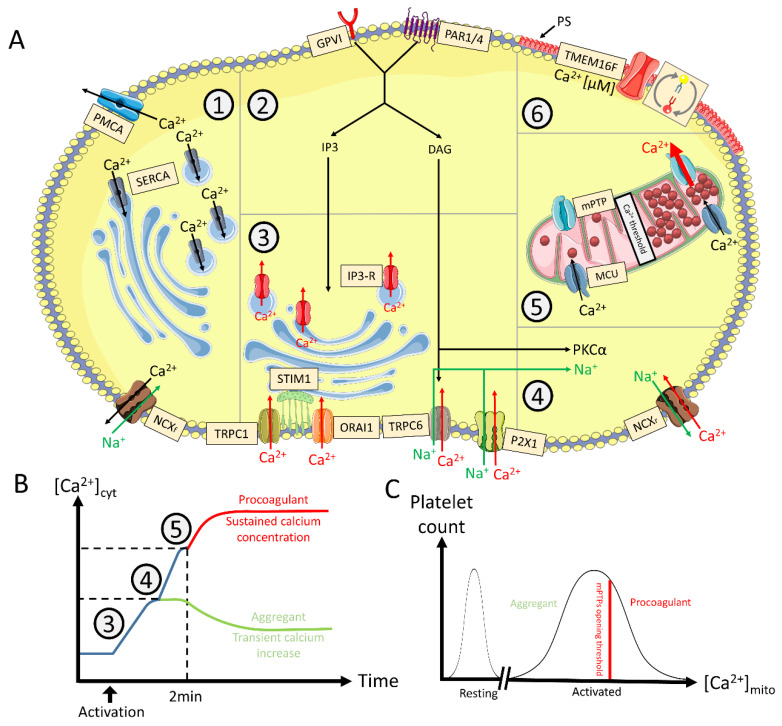
Calcium mobilization mechanisms and intracellular profile leading to procoagulant COAT platelets. **Panel** (**A**). **Cytosolic calcium regulation. *1. Downregulation***: The intracellular level of calcium (Ca^2+^) is negatively regulated (Ca^2+^ in black) by PMCA and the forward mode of NCX (Ca^2+^ efflux towards the extracellular space) respectively compartmentalized by SERCA in intracellular stores (granules, DTS) and MCU in mitochondria. ***2. Initial activation***: Engagement of GPVI (by collagen) and PAR1/4 (by thrombin) mediate IP3 and DAG production. ***3. Internal Ca^2+^ storage***: IP3 triggers the release of Ca^2+^ from the internal stores through its receptor while the stores are refilled by ORAI1 mediated by STIM1 (Ca^2+^ sensor). TRPC1 is either agonist or storage depletion dependent and increases cytosolic Ca^2+^ and sodium (Na^+^) when activated. TRPC6 activation is mediated by DAG. ATP released from granules activates the P2X1 receptor increasing cytosolic Ca^2+^ and Na^+^. ***4. NCX reversing***: Increased cytosolic Na^+^ and PKCα contribute to the reverse mode of NCX. ***5. Mitochondrial permeation***: Cytosolic Ca^2+^ is transferred into mitochondria through MCU. At a given threshold, and in some platelets, mitochondrial Ca^2+^ triggers mPTP opening, releasing an important amount of Ca^2+^ in the cytosol. ***6. PS exposure***: Sustained micromolar levels of cytosolic Ca^2+^ are necessary to activate low Ca^2+^-sensitive actors such as calpain and TMEM16F. The latter scrambles membrane phospholipids inducing the expression of PS in the outer part of the cytosplamic membrane. **Panel** (**B**). **Model of stepwise Ca^2+^ mobilization**. After initial [Ca^2+^]_cyt_ increase, some platelets do not trigger NCX reversing, and will remain aggregant decreasing their [Ca^2+^]_cyt_ (green). Other platelets undergo NCX reversing, thus further increasing [Ca^2+^]_cyt_. Platelets become procoagulant (red) approx. 2 min after activation, when Ca^2+^ concentrations in cytoplasm and mitochondria reach a threshold level to induce mitochondrial depolarization, supramaximal [Ca^2+^]_cyt_, and subsequent PS exposure. **Panel** (**C**). **Mitochondrial Ca^2+^ profile at the population level**. The level of accumulated mitochondrial Ca^2+^ depends on previous activation of the Ca^2+^ mobilization sources following platelet activation. Mitochondria of a platelet subpopulation accumulate enough Ca^2+^ to open their PTP and induce a procoagulant response in those platelets. The figure uses modified images from Servier Medical Art under a Creative Commons Attribution 3.0 Unported License (http://smart.servier.com, accessed on 31 January 2021). This figure is inspired from [22,26,27]. Legend: [Ca^2+^]_cyt/mito_: cytosolic/mitochondrial calcium concentration; DAG: diacylglycerol; DTS: dense tubular system; IP3: inositol trisphosphate; GP: glycoprotein; PAR: protease activated receptors; MCU: mitochondrial calcium uniporter; NCX_f/r_: sodium calcium exchanger forward or reverse mode, respectively; PMCA: plasma membrane calcium ATPase; PS: phosphatidylserine; mPTP: mitochondrial permeability transition pore; SERCA: sarco-endoplasmic reticulum calcium ATPase; STIM: stromal interaction molecule; TMEM: transmembrane; TRPC: transient receptor potential C.

**Figure 3 ijms-23-02536-f003:**
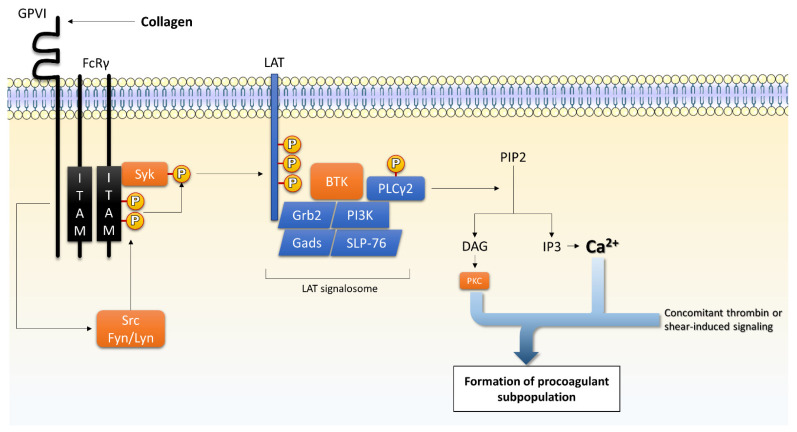
Collagen-mediated signaling via glycoprotein (GP) VI. In addition to adhesion mediated by GPIa-IIa (integrin α2β1), collagen induces platelet activation via GPVI. It triggers the formation of IP3 releasing Ca^2+^ from internal stores. PKC isoforms also participate in further platelet activation. This figure is inspired from [138,139,140]. Legend: BTK: Bruton’s kinases; Ca^2+^: calcium; DAG: diacylglycerol; FcRγ: Fc receptor γ-chain; IP3: inositol trisphosphate; ITAM: immunoreceptor tyrosine activation motif; LAT: linker for activation of T cells; MAPK: mitogen-activated protein kinases; PI3K: phosphatidylinositide-3-kinase; PIP2: phosphatidylinositol-4,5-bisphosphate; PLCγ2: phospholipase C γ2; Rap1: Ras-related protein 1; Syk: spleen tyrosine kinase; TXA2: thromboxane A2.

**Figure 4 ijms-23-02536-f004:**
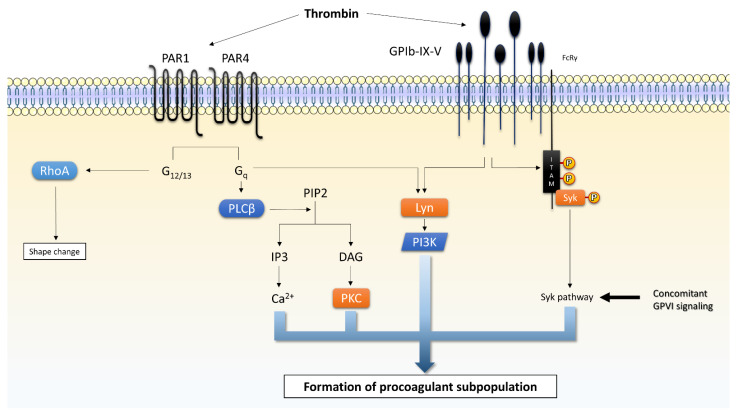
Thrombin-mediated signaling. Thrombin activates platelets through two types of receptors acting in a synergistic way: the G protein-coupled receptors PAR 1/4 and the GPIbα from the GPIb-IX-V complex. This figure is inspired, simplified, and adapted from [24,140,150]. Legend: Ca^2+^: calcium, DAG: diacylglycerol; G: guanine nucleotide-binding proteins; IP3: inositol trisphosphate; ITAM: immuno-receptor tyrosine activation motif; PAR: protease-activated receptors; PKC: protein kinase C; PI3K: phosphatidylinositide-3-kinase; PIP2: phosphatidylinositol-4,5-bisphosphate; PLC: phospholipase C; Rap1: Ras-related protein 1; RhoA: Ras homolog family member A; Syk: spleen tyrosine kinase.

**Figure 5 ijms-23-02536-f005:**
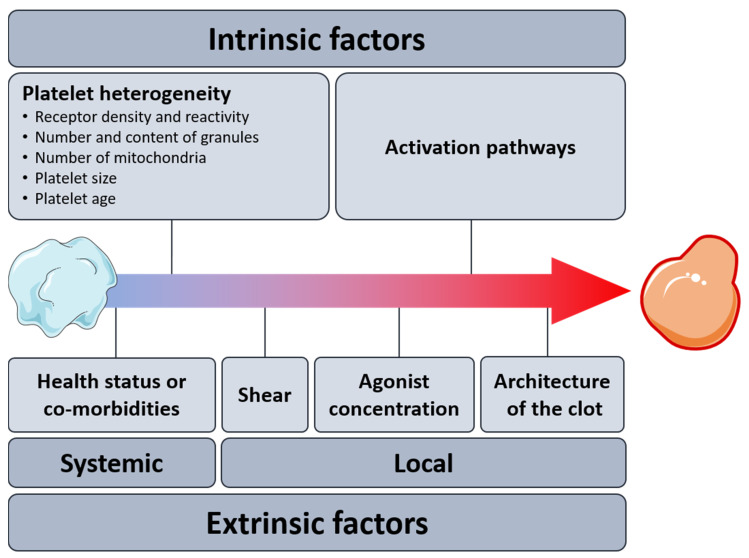
Intrinsic and extrinsic factors potentially driving the procoagulant response of platelets.

## Data Availability

Not applicable.

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
