# Peer review of "Mechanisms Underlying Dichotomous Procoagulant COAT Platelet Generation—A Conceptual Review Summarizing Current Knowledge"

_ijms, 2022, doi:10.3390/ijms23052536_

Round 1

Reviewer 1 Report

The review by Veuthy et al entilted Mecahnisms underlying dichotomous procoagulant COAT platelet generation - A conceptual review summarizing current knowledge is a comprehensive review covering the platelet signalling components driving procoagulant platelet function , with a key focus on Ca2+ signal transduction. While well written and highly comprehensive in scope, the review could be improved with some degree of focus. In its present form the review covers a lot of extraneous ground that could be cut while maintaining the core theme. The sections dealing with well trodden Ca2+ signalling pathways could be tightened for example.

Overall a useful review with some editing required

Author Response

IJMS-1599862 Response to Reviewers

Dear Editor

We thank the reviewers and the editorial board for their positive judgement and interest in our work. We are grateful for their fair comments and feedback, which helped us to improve the manuscript.

Here we provide point-to-point responses to the comments and suggestions.

Major changes are highlighted in “track change modus” in the revised manuscript.

Reviewer 1

The review by Veuthy et al entilted Mecahnisms underlying dichotomous procoagulant COAT platelet generation - A conceptual review summarizing current knowledge is a comprehensive review covering the platelet signalling components driving procoagulant platelet function , with a key focus on Ca2+ signal transduction.

Answer: We thank Reviewer 1 for the appreciation of our work.

1/1. While well written and highly comprehensive in scope, the review could be improved with some degree of focus. In its present form the review covers a lot of extraneous ground that could be cut while maintaining the core theme. The sections dealing with well trodden Ca2+ signalling pathways could be tightened for example.

Answer: We wrote this Review aiming at summarizing current knowledge on the mechanisms underlying procoagulant platelet formation for newcomers in the field. For this reason, if possible, we would keep the amount of information as it is. In particular, although several publications discuss calcium signalling in platelets, to the best of our knowledge, its sequential regulation in procoagulant COAT platelets and the dichotomous role of NCX have not been described elsewhere.

Reviewer 2 Report

The article by Veuthey et al provides a comprehensive review on procoagulant platelets, which includes their characterization and the underlying mechanisms that leads to the procoagulant state. The authors have done a remarkable job in covering the literature and making this article highly informative. However, recently there have been numerous articles that have been published on the same topic with similar content. Some minor comments are: 

  1. Re Page 33, Line 62-65: There is only a small proportion of the population in the growing thrombus that reaches the procoagulant state and the current antithrombotic strategies mostly target the aggregating platelets. How do the procoagulant platelets may have an impact on antithrombotics and how can they be considered while developing newer antithrombotics? 
  2. What are the clinical implications of Procoagulant platelets? This can be discussed in light of Scott syndrome and haemophilia A.
  3. Any recent advances in the development of anti-procoagulant platelet therapy can also be incorporated.
  4. Section 4 of the manuscript can be shortened by removing already well-known signaling pathways associated with different platelet receptors.

Author Response

IJMS-1599862 Response to Reviewers

Dear Editor

We thank the reviewers and the editorial board for their positive judgement and interest in our work. We are grateful for their fair comments and feedback, which helped us to improve the manuscript.

Here we provide point-to-point responses to the comments and suggestions.

Major changes are highlighted in “track change modus” in the revised manuscript.

Reviewer 2

The article by Veuthey et al provides a comprehensive review on procoagulant platelets, which includes their characterization and the underlying mechanisms that leads to the procoagulant state. The authors have done a remarkable job in covering the literature and making this article highly informative. However, recently there have been numerous articles that have been published on the same topic with similar content.

Answer: We thank Reviewer 2 for the appreciation of our work. If required, we will be happy to cite relevant publications not yet listed in our References.

Some minor comments are: 

2/1. Re Page 33, Line 62-65: There is only a small proportion of the population in the growing thrombus that reaches the procoagulant state and the current antithrombotic strategies mostly target the aggregating platelets. How do the procoagulant platelets may have an impact on antithrombotics and how can they be considered while developing newer antithrombotics? 

Answer: We have briefly addressed this point in the revised version of the manuscript (see page 3/33, lines 65-74).

2/2. What are the clinical implications of Procoagulant platelets? This can be discussed in light of Scott syndrome and haemophilia A.

Answer: We have briefly addressed this point in the revised version of the manuscript (see page 1/33, lines 40-44).

2/3. Any recent advances in the development of anti-procoagulant platelet therapy can also be incorporated.

Answer: We feel that this interesting question is outside the focus of our present review and we refer to our previous publication (PMCID: PMC7956450), cited as reference 2 in the revised version of manuscript, in which we discussed this point.

2/4. Section 4 of the manuscript can be shortened by removing already well-known signaling pathways associated with different platelet receptors.

Answer: See response to point 1/1 (see attached file)
